# Choosing Alternative Career Pathways after Immigration: Aspects Internationally Educated Physicians Consider when Narrowing down Non-Physician Career Choices

**DOI:** 10.3390/healthcare11050657

**Published:** 2023-02-23

**Authors:** Nashit Chowdhury, Deidre Lake, Tanvir C. Turin

**Affiliations:** 1Department of Family Medicine, Cumming School of Medicine, University of Calgary, Calgary, AB T2N 4N1, Canada; 2Department of Community Health Sciences, Cumming School of Medicine, University of Calgary, Calgary, AB T2N 4N1, Canada; 3Alberta International Medical Graduates Association, Calgary, AB T2E 3K8, Canada

**Keywords:** internationally educated physicians, alternative career pathway, career choice, decision factors, international medical graduates

## Abstract

Many developed countries admit internationally educated physicians (IEPs) as highly skilled migrants. The majority of IEPs arrive with the intention of becoming licensed physicians to no avail, resulting in underemployment and underutilization of this highly skilled group of people. Alternative careers in the health and wellness sector provide IEPs opportunities to use their skills and reclaim their lost professional identity; however, this path also includes great challenges. In this study, we determined factors that affect IEPs’ decisions regarding their choice of alternative jobs. We conducted eight focus groups with 42 IEPs in Canada. Factors affecting IEPs’ career decisions were related to their individual situations and tangible aspects of career exploration, including resources and skills. A number of factors were associated with IEPs’ personal interests and goals, such as a passion for a particular career, which also varied across participants. Overall, IEPs interested in alternative careers took an adaptive approach, largely influenced by the need to earn a living in a foreign country and accommodate family needs and responsibilities.

## 1. Introduction

Internationally educated physicians (IEPs) are individuals who have graduated from medical schools located outside of Canada [1]. They are also known as foreign medical graduates (FMGs), international medical graduates (IMGs), or internationally trained physicians (ITPs) [2,3,4]. Most IEPs are immigrants, refugees, or temporary migrants who moved to Canada after completing their medical graduation. Others are Canadian-born citizens who studied medicine abroad [1]. While these IEPs may have had successful medical careers in the countries they trained in, they often become un- or underemployed in Canada due to demanding and resource-intensive licensing procedures. Most IEPs cannot re-enter their intended career to become practicing physicians in Canada because of numerous individual and systemic barriers [5,6]. Generally, only 20–30% of IEPs are able to obtain a residency training spot, which restricts and hinders IEPs from obtaining physician licensure in Canada [7,8]. Many IEPs, thus, have become a symbol of the deskilling of highly skilled migrants in developed countries [9].

The low success rate of becoming a practicing physician in Canada causes frustration about life and career prospects in Canada, as well as financial strain leading, to being unable to bear family responsibilities [10,11]. This drives IEPs to look for viable alternate employment options based on their background [9]. In general, alternative careers are the “career options that immigrants pursue other than but related to the regulatory profession in which they were originally trained, that make use of and relate to an immigrant’s skills and experience” [5]. For IEPs, alternative careers may be defined as those jobs in the health and wellness sectors that utilize their medical skills and knowledge. A Canada-wide survey of 1740 participants found that 68.78% of employed unlicensed IEPs worked in health-related alternative professions, with the majority (50.45% of total employed) working in non-regulated professions [12]. However, a notable proportion (31.22%) were working in non-health-related fields [12]. As such, another nationwide survey of 356 unlicensed IEPs reported that the majority of the IEPs were dissatisfied (61%) with their current alternative professions and wished to have greater government and community support (93%) for the development of alternative careers for IEPs [11]. The IEPs also reported that their years of effort and exhausting of resources after licensing processes did not help in their current alternative profession as well [11].

The intent of an alternative career is not to start a career from scratch, but rather to find a position where IEPs can build on the education and training they already have. The non-recognition of all the qualifications and experiences of the IEPs and the lack of systemic support make the alternative career pursuit very challenging [5]. Less than 10% of IEPs currently employed in alternative careers accessed government-supported career-related training or education resources [11]. Having spent a good portion of their life training to become a physician, many IEPs have never considered doing anything else, are unsure of how their skills and knowledge base might be applied to other areas, and are uncertain how to prepare for alternative careers, including obtaining information and accessing resources [13]. Further, employers are unaware of this situation and IEPs’ potential for non-physician health and wellness-related jobs; they do not understand why a person trained to become a physician would want to do anything else [14]. The lack of information, organizational support, job counseling, and coaching further hinders IEPs’ pursuit of alternative careers [5].

The pathways to the professional integration of IEPs in Canada are already very complex, with multiple routes, requirements, and obstacles [15]. Learning about, deciding on, and ultimately pursuing potential alternative careers adds to that complexity. As such, studies have reinforced the urgent need to develop alternative career guides and supports [5,16]. While previous studies provided certain hints of the determining factors for IEPs (e.g., salary and further course requirements) in terms of choosing an alternative career came up as a part of the discussion focused on a different objective [5], none of these studies specifically examined the decision-making process of IEPs around alternative careers. Therefore, through this exploratory study, we planned to investigate the decision-making process of IEPs regarding their pursuit of alternative careers and aimed to identify factors IEPs need to consider when seeking alternative careers.

### 1.1. Prior Research and Significance of the Study

Canada continues to bring skilled migrants to health and other professional sectors to compensate for the aging population and low population growth [17]. Existing literature has discussed various issues of skilled migrants in general that are primarily driven by immigrant-serving organizations in collaboration with some academics, often those with immigrant or refugee backgrounds [18]. This includes un/underemployment of skilled migrants and the economic losses from this phenomenon [19], barriers and facilitators of credential recognition [20], work environment [21], perceived discrimination [22], and policies around skilled migration [23,24]. People with foreign-sounding names were 20–40% less likely to receive a call from human resources for job interviews [19], which forces them to be employed in low-skilled jobs for survival. Reitz et al. (2014) showed that deskilling causes $11 billion per year in economic losses in Canada. Moreover, this exploitation of skilled migrants affects their physical and mental health and adds more to the loss of human capital [25].

The issue is more prominent for regulated jobs. Augustine [26] compared regulations and policies for alternative routes to the primary careers of skilled migrants of three professions between Canada and Australia: accounting, engineering, and medicine. There are some acceptable alternative routes; however, they are not sufficient for the IEPs. This study reported that certain occupations such as physicians have more need for suitable alternative careers because their training is highly specific. While the use of supervised practice as physicians reduced the bottleneck for IEPs in Australia as opposed to Canada, however that may not result desirable outcomes in a competitive labour market. Alternative careers provide flexibility in integration of skilled immigrants into the labour market by allowing different ways to demonstrate competency or fill in gaps in education without redoing the whole degree or training. Professions like IEPs this is very crucial as in Canada most of the IEPs has to go through the full residency training to enter their primary profession [6].

There is scarce literature regarding alternative career pathways for IEPs in the health and wellness sector [16,27]. As one study noted, IEPs are on average 10 years older than Canadian medical school graduates and may have more obligations and responsibilities to their families [1]. They have smaller professional and social networks and greater financial constraints, which may affect their ability to successfully pursue a new career [28]. IEPs often wait too long to take steps toward alternative careers due to a lack of information, uncertainty, and fear of losing professional status [6]. While an alternative career is not what they have dreamed of and are prepared for, it offers them a way to reclaim their professional identity and earn a respectable living [29]. A study that interviewed IEPs and seven other professionals reported that their IEP participants suggested a toolbox that would help IEPs understand alternative job options and pursue them [5]. By identifying the thought processes of IEPs around alternative career pursuits, this study’s findings could be used to develop a tool that will enable IEPs to find suitable alternative careers and regain their professional identity and economic losses to a certain extent.

### 1.2. Theoretical Underpinnings

Human Capital Theory (HCT) has been the foundation of Canadian immigration policies in recent decades, which welcomes highly qualified migrant professionals based on their skill, education, and experience and intends to assimilate them back into their field of expertise following migration [30]. During the immigration process, most of them come in skill level category A or B (post-secondary advanced degree and experiences), determined by the Immigration, Refugees, and Citizenship Canada (IRCC). Unfortunately, contrary to the intent, the majority of them are forced to work in skill-level C or D jobs that do not need advanced degrees or expertise, such as taxi driving, retail and service industries, and so on. Moreover, employers of potential alternative careers often do not recognize IEPs’ educational qualifications and experience, preferring local Canadian credentials and experience. [31]. Paul and colleagues [32] identify that this scenario has been happening due to the disconnection between the policies of the federal, provincial, and local levels of governance. While at the federal level, policies recognize the educational training and experience of IEPs and welcome them into the country as skilled migrants, the provincial level often does not recognize their education and experiences [32]. This disconnection becomes worse at local levels where some employers fail to understand the level of skill and knowledge of the IEPs. Others, on the other hand, may reject them for non-physician health and wellness career (i.e., alternative) positions, considering them overqualified recognizing as IEPs, which is attributable to unawareness of the long, complex, and very uncertain physician licensure process for IEPs compelling IEPs to look for alternative careers. This complicated process results in the loss of recognition of the human capital of IEPs in the post-migration period [33,34]. It is important to work for the re-recognition of IEPs’ human capital in the post-migration period at all levels. In this study, we look into the IEP perspectives on pursuing alternative careers, which we believe is crucial to the development of any support programs for the facilitation of alternative career pursuits for IEPs.

We also draw on intersectionality theory in understanding the alternative career decision-making factors among IEPs. Intersectionality allows the understanding of phenomena through multiple and overlapping lenses of intersecting identities such as gender, race, socioeconomic, and immigrant origins [35]. The IEPs come from diverse socio-economic backgrounds and may have experienced a range of discrimination during their attempt to professionally integrate into Canada. Their socio-economic and individual determinants may influence their mechanism of the decision-making process around alternative careers. For example, women IEPs from certain cultural backgrounds may prioritize taking care of their families over a career and look for an alternative career with a more flexible work schedule or only part-time positions [8,36].

## 2. Methods

As this study focused on alternative career pathways for IEPs where there has been very little previous empirical work, we chose to adopt an exploratory qualitative descriptive approach to the methodology [37]. This study employed focus group discussions (FGDs) as a tool for data collection from IEPs. Due to COVID-19 restrictions at the time of data collection, the FGDs were conducted via Zoom, an online meeting software (Zoom Video Communications, Inc., San Jose, CA, USA), using a secured account provided by the University of Calgary. This also offered us the benefit of recruiting participants from across Canada.

All of the selected IEPs were either already working in an alternative career or had expressed an interest in pursuing an alternative career and had similar backgrounds in terms of being an IEP and having experience of facing myriad barriers to their career pursuit, which for the purposes of our study constituted a homogenous group. The barriers were explored by a different research question in the same setting, which has been published elsewhere [38].

### 2.1. Recruitment and Participants

Purposive sampling was used to recruit participants for our study. This is a non-probabilistic sampling where participants are selected by the researcher based on their knowledge, experience, and ability to expand on a certain topic, theme, or phenomenon [39]. Despite there being a chance of researcher bias, this sampling method is used in qualitative research to maximize the relevance of collected data to the research objectives [40].

Despite the chance of selection bias, we chose this method to ensure that only the correct participants (i.e., those who were really into an alternative career, not just having it as a thought) were recruited. Ethical approval was granted by the Conjoint Health Research Ethics Board (CHREB) at the University of Calgary.

Our study included a total of 42 participants across eight focus groups. Most participants were either Canadian citizens (naturalized) or permanent residents, and only two were temporary residents (Table 1). A total of 25 participants were employed and most of them (20 participants) worked in health-related alternative careers. Among the 17 unemployed participants, 14 were actively seeking work at the time of the focus group discussion. Most of the participants were female (31 vs. 11), and most belonged to the age groups 30–39 and 40–49 years (17 and 14 participants, respectively).

### 2.2. Data Collection

Each of the eight focus groups comprised four to seven participants. The FGDs were facilitated using a semi-structured questionnaire that was developed by the research team and reviewed by two IEP citizen researchers. Each FGD was audio-recorded using Zoom recording options and transcribed verbatim. All FGDs were conducted in English. Each session lasted from one hour to one and a half hours.

### 2.3. Data Analysis

We adopted an inductive thematic analysis approach to analyze the data [41]. We exported the transcriptions to NVivo qualitative data analysis software (QSR International, Version 12, Melbourne, VIC, Australia), which was used to generate codes and themes. At first, NC coded the data from the transcription of the first three FGDs and then met with the other members of the team to examine the coding for appropriateness and biases. Following a discussion with the team, the initial coding scheme was rectified. NC then continued coding the rest of the focus group transcripts, and the team reconvened after completion of the coding of all eight focus groups (which yielded 21 initial codes). The team discussions led to the finalization of all codes, followed by sub-themes and themes. The validity of the data was determined by several procedures, including through the lens of the researcher, the study participants, and IEPs external to the study. In addition, a representative sample of the participants member-checked the quotes and the findings.

## 3. Results

The analysis of the focus group discussions revealed five themes regarding determining factors IEPs consider while exploring alternative career options. The first theme, “Qualification and experience requirement”, included the requirement of a certain number of hours of Canadian voluntary or paid work experience, certification, and certain undergraduate or postgraduate degrees/diplomas. The second theme was “Personal resource requirement for capacity building”, which entailed the time and cost required to be competitive for a particular job. The third theme, “Possibility of the utilization of transferable skills”, included factors related to the extent of transferability of the skills earned in medical schools and experience as physicians. The fourth theme, “Employment-related factors”, represented those factors that arise from being employed in alternative careers, such as financial compensation, opportunities for further growth within the job, and others. The final theme, “Personal factors”, addressed intangible factors arising from personal preferences and constraints.

### 3.1. Theme 1: Qualification and Experience Requirement

#### 3.1.1. Sub-Theme: Qualification Requirement

Participants indicated that one of the first things they would consider in choosing an alternative job was the qualifications required for that job.

“Yeah, sure. I’m actually very, um, enthusiastic about learning. So I am doing the clinical research certificate and I want to work and I don’t mind going back to school. I really want to be settled here, so I don’t mind going back to school.”(FGD4P5), FGD: Focus Group Discussion; P5: Participant No. 5.

However, their views differed regarding the need for the qualification requirement. While some were happy to take up further training/institutional education for an alternative career, others were not interested in further schooling. Some would consider additional training, but they would prefer programs that are quick and easy and that complement their backgrounds.

“Uh, the thing that I am not open to is, as I mentioned before that I am at this point. Yeah. I’m not looking into going to school or going to upgrade myself. Uh, I think that it’s, it’s not in me. I lost that desire to do it. So I’m looking for something that can give me an opportunity to work now, instead of asking me to go back to school for one or two years, um, the otherwise I’m, I’m, uh, I can’t think of anything in particular. Uh, other than that.”(FGD4P2)

A few participants also mentioned that they would look into whether an English language proficiency test (such as the International English Language Testing System (IELTS)) and/or other credential evaluation tests (such as the Graduate Record Examinations (GRE)) was required as prerequisites to pursue further education for alternative careers.

“So, so first off for the preparatory phase, I mean to say that certain courses require extensive qualification, uh, for example, um, like IELTS score GRE score experiences. Uh, so just to get into that course, I think that, yeah.”(FGD8P3).

#### 3.1.2. Sub-Theme: Experience Requirement

Participants also said that they tried to narrow down their career choices based on the probable importance of Canadian volunteer or paid work experience for that specific type of job. Many participants had the perception that employers preferred the Canadian experience and did not recognize their extensive experience from back home or in other countries.

“You actually have the skills, you have the certification, but they still want you to have had the Canadian experience, which still makes it tedious. Even with this medical office administration. The fact that I even went to school here, they still want me to have had two years experience.”(FGD3P3).

### 3.2. Theme 2: Personal Resource Requirement for Capacity Building

#### 3.2.1. Sub-Theme: Time Requirement

The time required to become prepared for a job by earning a degree, going through training, and accumulating volunteer experience was voiced by many participants. Participants were keen to develop and be eligible or competitive candidates for a career, either by taking short-term training (e.g., 6 months to 1 year) or enrolling in programs that were often two or more years in length.

“I will think about the time factor. So I, um, I can take any program or any, uh, on any course for a short time period. Um, maybe up to a year or two years maximum, uh, until I finished my studying and exams on so much, but more than that, I think it.”(FGD4P6).

Length of time varied according to their personal choices and interest and was influenced by their family responsibilities, financial circumstances, and career goals.

#### 3.2.2. Sub-Theme: Cost of Required Courses

Participants also recognized that certain career pathways might require them to undergo additional courses or retraining. The cost related to this capacity was also considered a factor in their decision of an alternative career.

“…you know, first and foremost, what I think is, uh, what is the cost of that course? Have to do, if I have to do a course and, uh, if I need, you know.”(FGD1P1)

However, the cost might be compared to the potential outcome of the job in terms of job certainty and financial outcome.

“That’s not a problem for me, if it is needed, I’m also. Ready to invest for a course or whatever, but really the ever-changing Canadian and [muffled] landscape will be able to provide me with a job once I come out of it.”(FGD1P1)

### 3.3. Theme 3: Possibility of the Utilization of Transferable Skills

#### 3.3.1. Sub-Theme: Skills from Previous Specialty or Additional Professional Experience

IEPs in general showed interest in finding an alternative job that would utilize their clinical skills and knowledge. IEPs looked for these jobs because they felt it would require less effort and fewer additional resources for them to obtain the required knowledge and skills for those alternative jobs that suited the medical field.

“Third thing, uh, trying to select something, that near to the medical field. Okay. So that it will not take lots of money from the person who is trying, for example, to study or to prepare to get in that field. And it will not be time-consuming the same time.”(FGD8P4)

Another purpose of participants looking for careers utilizing clinical skills and knowledge was to ensure continuous use of such skills, which would ultimately benefit participants when pursuing medical licensure in Canada in the long term, while pursuing alternative careers to meet immediate goals and needs. This was reiterated by those participants who had specialty training or additional qualifications such as a certification in Diagnostic Sonography. Participants intended to use these qualifications and experiences to guide their career search to a position where they could employ such experiences.

“…since I have done the ARDMS (American Registry for Diagnostic Medical Sonography). So, my first priority will be, when I think about the alternative career, it will be something related to this field. Like radiology or something, I know that’s really hard, but that will be my first priority and something related to patient.”(FGD2P6)

#### 3.3.2. Sub-Theme: Non-Specific Transferable Skills

Some participants pointed out that as a physician they also had experience with certain essential skills such as managing, interpersonal skills, communication skills, and administrative skills. They would also consider these when shortlisting their alternative career choices.

“So I have been working with people, managing people, a part of work that I have done before. Also, it has given me those skills to work with. So. People rather than just dealing with patients.”(FGD4P2)

### 3.4. Theme 4: Employment-Related Factors

#### 3.4.1. Sub-Theme: Wages or Financial Outcome

The salary/wages of jobs were a crucial factor for the pursuit of alternative careers, which was a common theme that arose from the discussions.

“So financial outcome. Yes, remuneration. Actually, what you’re going to get, it should be sufficient enough to maintain dignity as well as your life.”(FGD1P3)

Most participants wanted a career that paid a decent amount; however, depending on other factors, such as growth opportunity, they might waive the priority of financial gain.

“Yeah, top three for me would be one would be income, um, location and three would be, um, how close is it to my actual work? So income, location, and the third one, the closeness to my role as a doctor.”(FGD4P3)

#### 3.4.2. Sub-Theme: Flexibility in Working Hours

Many participants were concerned about the working hours for the job. It was often expressed as a surrogate for flexibility. Some participants were interested in part-time jobs to give more time to their families or have time to study for Canadian medical licensure and other long-term career options.

“That I might hold some kind of job, which is not too hectic to have a family and, um, to, to, to give time to the husband, to the kids. And you know, it’s not like, uh, a difficult, you know, like, well, flexible hours relaxing, but it’s still some, you know, some kind of income, basic income.(FGD2P5)

“…and I consider that is, working hours. Do I have to work on weekends? I have to do nights, or it’s just weekdays?”(FGD4P3)

#### 3.4.3. Sub-Theme: Opportunity for Growth

Another important employment-related factor was the opportunity for growth and career progression within an alternative career. Some participants expressed not wanting a job with no opportunity for promotion to higher positions accompanied by salary raises.

“I needed to be in a, in a profession or something that I would like, can you going on and what I can grow and just continue and feel satisfied with it.”(FGD3P2).

When asked to elaborate on how participants would know if the job had growth opportunities, participants mentioned they would explore to see whether the position had mentorship or job coaching programs, which may be indicated in the job description.

“Yep. Um, so for the growth opportunity, I would say, you know, it might not be even again, a tangible thing that a lot of time actually come around with assistant development opportunities are provided or, you know, a mentorship is provided or coaching job coaching is private or something like that.”(FGD8P5).

One participant, however, suggested that information on growth opportunities was not usually found to be documented, and participants had to communicate with individuals already employed by the organization to explore this aspect of the position.

“Initially they said that there was no opportunity for growth, but then as I was in the job, I saw a lot of opportunities as I went through it. Cause like, um, I had the opportunity to be part of research studies as well as our medical lab technician. So I was in contact with a lot of physicians, like in Alberta Children’s Hospital. And then at the same time now at the South health campus, we do a lot of research studies. And then there’s also this opportunity for more of the administrative part, like knowing the ins and outs of the laboratory, how it works.”(FGD5P5).

#### 3.4.4. Sub-Theme: Job Demand and Availability

Participants reported that they often explored the market demand and long-term outlook of a career before pursuing it. This was especially the case when educational training was required, where participants were more concerned about the availability of jobs within and the sustainability of a career once they had invested their time, effort, and money to become qualified.

“Cause whenever I want to train on something, I want to train it good, like a hundred percent perfection, but the money that is required to do the training, if it has to come from me and I would want to have some kind of, um, reassurance that I’m going to get a job at the end.”(FGD2P5).

#### 3.4.5. Sub-Theme: Networking Opportunity

Some participants indicated that they would investigate whether there was a networking opportunity within an alternative career. To elaborate, participants mentioned that they would look to see whether there was an opportunity for collaborative work with other institutions or organizations so that the professional network could be widened.

“Um, yeah, many of the research groups that I’ve known I’ve worked with or know about they work in collaborations. So there’s a lot of opportunity to work with different stakeholders and different collaborators, even though they are different than universities that are collaborating together on a project.”(FGD6P1).

One participant mentioned that working in a team environment might be a good opportunity to widen their personal professional network.

“So if you’re working in a team environment, you’re working with like, you know, 10 or 15 different peoples, you will get to know them. There is a networking opportunity, for sure.”(FGD8P2)

However, participants did feel that the ability to successfully network was largely due to the personal competencies of the IEPs to build and maintain professional relationships.

“Yeah. You can’t give it right. Like you can’t dictate it. Uh it’s because honestly, um, but from person to person, the networking ability differs.”(FGD8P1)

#### 3.4.6. Sub-Theme: Work Environment

When seeking jobs, participants prioritized workplaces that possessed a collaborative attitude, gave importance to stress management, had effective leadership from management, and had good staff benefits and support.

(From the Zoom chat feature) “Collaborative work environment, amount of stress and what do the employer (do) to manage staffs’ stress. One of the proxy could be what proportion of people are working in the specific workplace for what duration. How much they appreciate their staffs for the work they do. Paid vacation, sick leaves, family leaves, paid holidays, etc.”(FGD8P5).

### 3.5. Theme 5: Personal Level Factors

#### 3.5.1. Sub-Theme: Passion or Interest for Certain Jobs or Work Area

Certain participants mentioned that the alternative career had to be something that was meaningful to them and satisfied them as an individual. Participants elaborated to say that careers should be something that involves patient engagement and care and decision-making capacity regarding patient care, something that is an important attribute in practicing physicians.

“Like radiology, or something, I know that’s really hard, but that will be my first priority and something related to patient. Actually, since I, I worked solely through the clinical side. I don’t know. Is it possible or what, but if I got chance to work with the patient, uh, I will, I will definitely go through that, that being my first priority.”(FGD2P6)

#### 3.5.2. Sub-Theme: Alignment with Future Goals

Some participants mentioned that the job must relate to their personal future goals, which varied among participants. For example, some IEPs who had the intention of pursuing Canadian medical licensure focused on careers that would ultimately benefit their chances of gaining residency spots when going through the licensure process.

“…either that or that particular job is going to be a bridging job. Uh, or a position like a stepping stone, uh, for me to, uh, work for a little bit and then move on to something that is going to be my destination.”(FGD4P4).

## 4. Discussion

### 4.1. Expositions

Our study’s findings indicate that IEPs interested in alternative careers took an adaptive approach in general, perhaps influenced by the need to earn in a foreign country and accommodate family needs and responsibilities. Many interrelated factors and barriers, from very individualized (e.g., passion for a certain career) to very general (e.g., wages/salary) to systemic (e.g., recognition of foreign education/experience), are associated with IEPs choosing alternative careers in combination with the resettlement challenges they face in Canada. The IEPs go through a critical process of finding a career that balances all these factors and constraints.

Non-recognition of immigrants’ education and work experience has been well documented in the literature, and it results in limited access to opportunities IEPs can pursue with interest, passion, and dignity [42]. These experiences were also shared by our participants and other studies and deemed as systemic discrimination, including ageism and racism [43,44]. The results of our study especially shed light on the significant training and skills required to enter certain alternative careers for IEPs, causing IEPs to spend resources to update their skillset to match job trends within Canada [45]. These decisions to update skills, including meeting certain score requirements in English proficiency tests often require immense resources, such as time and money, leading to a disproportionate rate of unemployment, underemployment, and lower earning for IEPs as compared to the native Canadian professional population [46]. Further, many IEPs present to Canada with significant international experience; however, studies have shown the systemic disadvantage skilled immigrants face when compared to other professionals with Canadian experience [47].

A study that explored alternative careers for certain internationally trained professionals, including IEPs, concluded that IEPs are the driver in making the decision as to which alternative careers to pursue based on their backgrounds, which was reflected in this study [5]. Moreover, identifying one’s transferable skills was noted as a key element in successful transitions to alternative professions. We also observed that the participants were seeking careers where they could employ their transferable skills. Likewise, Peters et al. (2011) in Ontario reported that IEPs’ experiences mimic the responses of IEP participants in our study [48]. Further, a study by Sood et al. (2020) pointed out that IEPs may not necessarily secure an alternative job even after receiving training in Canada and investing their financial capital in it and may need to switch careers again [10]. Worries regarding this sort of outcome were also frequently mentioned by the participants in our study.

The results of this study also highlight the importance of both objective and subjective career success as important determinants of choosing alternative careers. Apart from IEPs utilizing their human capital, their personal feelings of self-accomplishment and pride, or subjective career success, are an important facet of transitioning into a successful contributing member of society [49]. Participants in this study commonly linked job satisfaction with not only growth opportunities but also passion and interest in the job. Participants focused on certain jobs where they can be satisfied intellectually and have an interest and passion in the area. However, passion often may not coincide with the transferable skills and experience they require to achieve an alternative career, causing a mismatch in expectations and reality [50].

This study presents an exploration of important factors in career exploration for IEPs and may assist in the development of a career decision-making tool specific to IEPs and assist them in finding alternative careers. Prior studies that discussed the career decision-making model match the findings of our study [51]. Scholars have discussed models of career decision-making that include consideration of rational factors focused on maximizing individual gains, such as financial outcome, career growth, and future goals, as well as other less rational factors that include a passion for certain jobs, individual satisfaction, and perspectives [52]. Our study findings revealed both kinds of factors in the alternative career decision-making process, which is drawn from a multicultural perspective and has high potential to be effective [53]. Likewise, another study compared three career decision-making strategies for employees going through a career change that included rational (based on careful thought), intuitive (based on emotional satisfaction), and dependent (assistance/approval from others) strategies, finding that a combination of all three, especially using the head (rationality) and heart (intuition), makes the most effective career choices [54]. Figure 1 shows how the decision-making factors identified in our current study can be integrated into these three strategies for adaptive career decision-making by IEPs.

### 4.2. Limitations

The moderator of the focus groups ensured that all participants contribute to each idea to reduce bias in our findings due to dominant participants. Despite the lack of interpersonal interactions, conducting focus groups online came out as rather beneficial for this study as the participants could express themselves more freely and could join from their homes anywhere in Canada [55]. However, most of the participants in this study were from Alberta (71.4%) due to our purposive sampling technique. We wanted to make sure that the participants were genuinely working in or considering alternative careers, which was more feasible for us from Alberta compared to other provinces. Further, perhaps participants from Alberta felt more interested than those from other provinces in participating in research conducted by the local and familiar university and community organizations. As IEPs across Canada encounter similar struggles in achieving their primary career and have similar educational and socio-cultural backgrounds, we believe the findings of our study can be applicable to a great extent in other provinces. We also observed a higher proportion of female participants in our study. We acknowledge this might be a limitation due to our chosen non-random sampling technique; however, other population-based studies also found a higher proportion of female participants in their studies [11,12].

### 4.3. Implications

This study extracted the thoughts and views of IEPs on alternative career choices, which can help inform future research and professional integration. The findings of the study can be used to develop a decision-support tool for IEPs who are considering alternative career options [56]. The research team aims to develop a web-based and/or mobile device application tool to help guide IEPs in choosing an alternative career according to their interest, skills, and other individual factors related to entry into and outcomes of the job. Moreover, the perspectives of potential employers, institutions that offer training or courses for alternative careers, and other stakeholders need to be captured and integrated to facilitate IEPs’ pursuit of an alternative career and remove unconscious bias. A concept note outlining the potential strategies and future research recommendations was published to inform policymakers, researchers, service providers, and other stakeholders [57]. Furthermore, IEPs in different professional roles (trans-professional adaptation) need to be evaluated [58].

The integration of IEPs and other skilled immigrants into the Canadian economy is a major policy issue within Canada and can have a significant impact on local economies. Utilizing the human capital that IEPs bring to their host countries can have major implications not only for the career success of economies, but also for major workflows, policies, and labour shortages, especially during a time of globalization when economies welcome international expertise. As seen through this study, IEPs use a series of contextual and personal factors to determine alternative career choices, making them self-reliant and proactive in their career behaviours. This career self-management is an excellent opportunity for labour policymakers and employers to assist IEPs through supportive human resource policies that focus on development, training, teamwork, and experiential learning that can assist IEPs to transition successfully into the labour market. This type of career self-management should be supported by labour policy to increase both the objective and subjective career success IEPs may gain from their alternative careers.

## 5. Conclusions

This qualitative study explored the decision-making process of IEPs pursuing alternative careers. A variety of influencing factors was identified that arise from educational qualification and professional experience, capacity-building resources, transferable skill utilization, employment, and individual-level concerns and priorities. These can be used to develop a decision-making support tool for IEPs considering alternative careers. Considering the difficult challenges IEPs encounter after moving to Canada and failing to succeed in their natural order of career—to become a physician—it is imperative to work toward supporting them to find and grow in suitable alternative careers and redress systemic barriers. While these factors and the hypothesized decision-support tool will help facilitate alternative career choices for IEPs, research and engagement initiatives to inform employers and other stakeholders about the IEPs’ situation and potential strategies to enhance their professional integration through alternative careers should be recognized as a pressing need.

## Figures and Tables

**Figure 1 healthcare-11-00657-f001:**
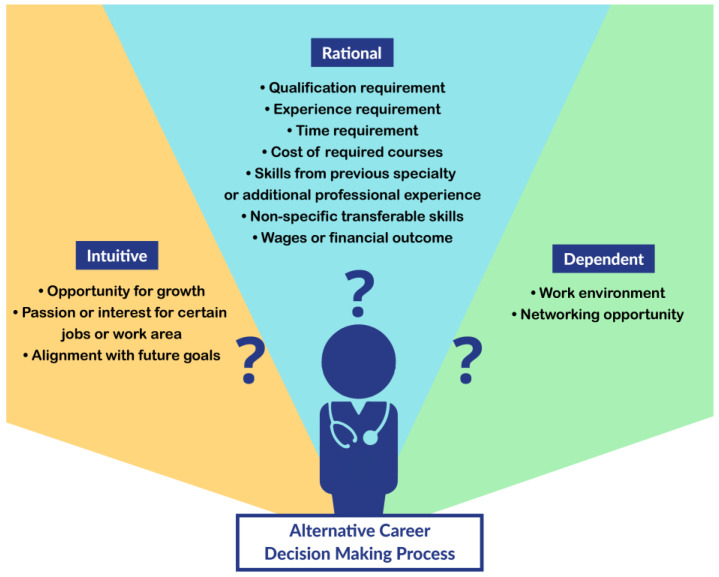
The alternative career decision-making process for IEPs.

**Table 1 healthcare-11-00657-t001:** Characteristics of study participants.

Traits	%	Count
Age		
29 or younger	9.5	4
30–39	40.5	17
40–49	33.3	14
50 or over	16.8	7
Sex		
Male	26.2	11
Female	73.9	31
Country of origin		
Armenia	2.4	1
Bangladesh	12.0	5
Canada	4.8	2
China	2.4	1
Colombia	2.4	1
Egypt	2.4	1
India	14.3	6
Iraq	2.4	1
Mexico	2.4	1
Nepal	2.4	1
Nigeria	14.5	6
Pakistan	21.4	9
Philippines	7.1	3
Somalia	2.4	1
Spain	2.4	1
Sudan	2.4	1
United Kingdom	2.4	1
Province currently living in		
Alberta	71.4	30
British Columbia	7.1	3
Manitoba	4.8	2
Ontario	14.3	6
Quebec	2.4	1
Immigration status		
Citizen	47.7	20
Permanent resident	47.7	20
Refugee	0.0	0
Temporary migrant (on a student visa, work visa, or visitor visa)	4.8	2
Specialty before coming to Canada		
Emergency medicine specialist	4.8	2
Family/general physician	38.1	16
Nephrologist	2.4	1
Neurological surgeon	2.4	1
Obstetrician	7.1	3
Occupational medicine specialist	2.4	1
Ophthalmologist	4.8	2
Paediatrician	4.8	2
Radiologist	4.8	2
Surgeon	2.4	1
Other	26.2	11
Others included: MPH, MD, FCPS, FRCS, or other post-graduate training in various specialty		
Current work position		
Employed (full-time)	33.3	14
Employed (part-time)	26.2	11
Unemployed; seeking work	33.3	14
Unemployed; not seeking work	7.1	3
Current area of work (among employed 25 participants)		
Health-related (regulated alternative career, i.e., requires licensure procedure, e.g., nursing, pharmacy technician, EMS tech, sonography, or laboratory technician)	20.0	5
Health-related (non-regulated alternative career, i.e., does not require licensure, e.g., health educator, health administrative officer, researcher, health policy analyst)	60.0	15
Non-health-related professional job (non-medical career build-up, e.g., engineering, business, or life sciences)	8.0	2
Non-health-related non-professional job (i.e., survival job, e.g., Uber/taxi driving, store jobs, or business owner)	12.0	3
Years spent preparing for alternative careers		
Less than a year	35.7	15
1–3 years	47.6	20
4–5 years	9.5	4
More than 5 years	7.1	3

## Data Availability

The datasets generated and analyzed during the current study are not publicly available due unavailability of the participants’ permission, privacy considerations, and ethical restrictions. Code-level deidentified data might be available from the corresponding author on reasonable request after all the planned publications are completed.

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
