# Peer review of "Choosing Alternative Career Pathways after Immigration: Aspects Internationally Educated Physicians Consider when Narrowing down Non-Physician Career Choices"

_healthcare, 2023, doi:10.3390/healthcare11050657_

Round 1

Reviewer 1 Report

The file is in the attachment.

Author Response

Comments from Reviewer 1

1. The paper proposed contributes to the issue of migrations of highly skilled migrants, a consolidated topic in migration studies and, specifically, of those Internationally Educated Physicians, who satisfy the demand of these professionals in developed countries as Canada.

It offers a new perspective regarding the pathways these migrants must overcome to get the position of physicians, analyzing their integration in other activities in the sector of health and wellness and the personal challenges that it represents.

Reply:
Thank you for your appreciating comments.

2. Theoretical framework: The Human Capital Theory and the Intersectionality Theory has been properly selected as the study lens of the phenomenon analyzed.

Reply:
Thank you for your acknowledgement of appropriate theoretical framework.

3. Methodology: The qualitative approach has been well chosen to this type of study. The selection of a focus group discussion was a good selection to overcome the difficulties inherent to the limitations of mobility during the pandemic. Perhaps, in the future, it would be necessary to increase the sample involving IEPs from other regions of Canada or even from other countries to develop a comparative analysis.

Reply:
Thank you for the positive comment on methodology and suggestions for the increasing sample from other regions of Canada/other countries. We believe that would be very useful and will consider for our future studies on the topic.

4. Results: The selection of the themes analyzed has been well approached.

Reply:
Thanks for the comment.

5. Discussion: The most valuable is the proposal to consider the necessary combination of two models of career decision-making process (rational factors and emotive factors) in the interpretation of the policies that must guide a successful integration of IEPs in Canada.

Reply:
Thank you for your diligent observation and comment.

6. Conclusions

The reflections on a decision-making support tool appropriate to facilitate their goal of working as physicians are the most interesting contribution of the paper.

Reply:
Thank you for your recognition of the impact of this study.

7. Weakness of this paper: In Table 1. Characteristics of study participants, in the section Country of origin two participants from Canada are counted. This is a contradiction with the content of the research given that strictly speaking they are not immigrants. These two IEPs must be excluded or their incorporation to the study must be clarified.

Reply:
Thank you for your constructive comment. IEPs are not exclusively immigrants. Some may be born in Canada and study medicine abroad and then come back again in Canada. They also face the similar (perhaps to a lesser extent given being from the same language and culture) barriers and challenges in integrating into Canadian labour market both in their original professions and alternative careers. Therefore, we intend to include them as well in the studies.
We also clarified that in the introduction of the revised manuscript: Page 1, lines 22-32

“Internationally educated physicians (IEP) are individuals who have graduated from medical schools located outside of Canada [1]. They are also known as foreign medical graduates (FMGs), international medical graduates (IMGs), or internationally trained physicians (ITPs) [2–4]. Most IEPs are immigrants, refugees, or temporary migrants who moved to Canada after completing their medical graduation. Others are Canadian-born citizens who studied medicine abroad [1]. While these IEPs may have had successful medical careers in the countries they received their medical degree and/or training, they often become un- or underemployed in Canada due to demanding and resource-intensive licencing procedures. Most IEPs cannot re-enter their intended career to become practicing physicians in Canada because of numerous individual and systemic barriers [5,6].”

8. Figure 1 must be better designed.

Reply:
Thank you for the suggestion. We were taking professional help to design the Figure. But could not make that prior to this re-submission due to deadline. Hopefully, we will be able to add the Figure at the next stage. Just for your kind note, the graphics will be updated, the contents will remain same.

Reviewer 2 Report

Thank you for allowing me to review the article "Choosing alternative career pathways after immigration: Aspects internationally educated physicians consider when narrowing down non-physician career choices”.

This study focused on alternative career pathways for IEPs.

The article is interesting but some problems do not allow the article to be published in its current version.

First of all, I trace a lack of review of the literature on the topic and a discussion of the results in light of it. It is important to describe the previous literature on alternative career choice in this population.

Related to the methodology used, the authors conducted eight focus groups with 42 IEPs in Canada. It is necessary for the authors to better specify sampling procedures, focus group coding, and the agreement among coders.

Finally, the results open up important practical implications. Practical/social implications should be discussed.

Author Response

Comments from Reviewer 2

  1. Thank you for allowing me to review the article "Choosing alternative career pathways after immigration: Aspects internationally educated physicians consider when narrowing down non-physician career choices”. This study focused on alternative career pathways for IEPs.

Reply:

Thank you for your kind comment.

  1. The article is interesting but some problems do not allow the article to be published in its current version. First of all, I trace a lack of review of the literature on the topic and a discussion of the results in light of it. It is important to describe the previous literature on alternative career choice in this population.

Reply:

Thank you for this comment. It is a new frontier in research therefore; there is only so much literature available on alternative career this topic. We previously drew on all the available literature on this topic available then. Nevertheless, we have included the information from a couple of very recently published articles.

See page 1-2, lines 36-58

"The low success rate of becoming a practicing physician in Canada causes frustration about their life and career prospects in Canada as well as financial strain leading to being unable to bear family responsibilities [10,11]. This drives IEPs to look for viable alternate employment options based on their background [9]. In general, alternative careers are the “career options that immigrants pursue other than but related to the regulatory profession in which they were originally trained, that make use of and relate to an immigrant’s skills and experience” [5]. For IEPs, alternative careers may be defined as those jobs in the health and wellness sectors that utilize their medical skills and knowledge. A Canada-wide survey of 1,740 participants found that 68.78% of employed unlicensed IEPs worked in health-related alternative professions, with the majority (50.45% of total employed) working in non-regulated professions [12]. However, a notable proportion (31.22%) were working in non-health-related fields [12]. As such, another nationwide survey of 356 unlicensed IEPs reported that the majority of the IEPs were dissatisfied (61%) with their current alternative professions and wished to have greater government and community support (93%) for the development of alter-native careers for IEPs [11]. The IEPs also reported that their years of effort and exhausting of resources after licensing processes did not help in their current alternative profession as well [11].

The intent of an alternative career is not to start a career from scratch, but rather to find a position where IEPs can build on the education and training they already have. The non-recognition of all the qualifications and experiences of the IEPs and the lack of systemic support make the alternative career pursuit very challenging [5]. Less than 10% of the IEPs currently employed in alternative careers accessed government-supported career-related training or education resources [11]. " 

  1. Related to the methodology used, the authors conducted eight focus groups with 42 IEPs in Canada. It is necessary for the authors to better specify sampling procedures, focus group coding, and the agreement among coders.

Reply:

Thank you for your suggestions. We have clarified the sampling and the coding process in the revised manuscript.

Please see page 4, lines 176-171

2.1. Recruitment and Participants

Purposive sampling was used to recruit participants for our study. This is a non-probability sampling where participants are selected by the researcher based on their knowledge, experience, and ability to expand on a certain topic, theme, or phenomenon [35]. Despite there being a chance of researcher bias, this sampling method is used in qualitative research to maximize the relevance of collected data to the research objectives [36]

And, page 6, lines 195-203

2.3. Data Analysis

We adopted an inductive thematic analysis approach to analyze the data [37]. We exported the transcriptions to NVivo qualitative data analysis software (QSR International, Version 12, Melbourne), which was used to generate codes and themes.

At first, one coder, NC coded the data from the transcription of the first three FGDs and then met with the other members of the team to examine the coding for appropriateness and biases. Following a discussion in the team the initial coding were rectified. NC then continued coding the rest of the focus group transcripts and the team reconvened after completion of the coding of all eight focus groups (which yielded 21 initial codes). The team discussions led to the finalization of all codes, followed by sub-themes, and themes. The validity of the data was determined by several procedures, including through the lens of the researcher, the study participants, and IEPs external to the study. In addition, a representative sample of the participants member-checked the quotes and the findings.

  1. Finally, the results open up important practical implications. Practical/social implications should be discussed.

Reply:

Thank you for the comment. We have updated the implications section to accommodate your suggestion.

Please see page 13, lines 493-503

“The findings of the study can be used to develop a decision-support tool for IEPs who are considering alternative career options [56]. The research team aims to develop a web-based and/or mobile device application tool to help guide IEPs in choosing an alternative career according to their interest, skills, and other individual factors related to entry to and outcomes of the job. Moreover, the perspectives of potential employers, institutions that offer training or courses for alternative careers, and other stakeholders need to be captured and integrated to facilitate IEPs’ pursuit of an alternative career and remove unconscious bias. A concept note outlining the potential strategies and future research recommendations was published to inform policymakers, re-searchers, service providers, and other stakeholders [57]. Also, IEPs in different professional roles (trans-professional adaptation) need to be evaluated [58]. 

Reviewer 3 Report

This is a paper on an extremely important topic that applies to a number of countries like Canada - the problem appears to be that medical qualifications and registration systems that are not integrated with migration policies - a systemic problem because the medical people insist it is about patient safety and not elitist views of the local qualifications, whereas the immigration policy folks place emphasis on level scores rather than content - an example of two silos failing to talk to each other. Useful point: Canadian-born/foreign educated have the same problem (albeit with a sample of only 2 people and from the paper we do not know whether they are male or female or what their specialities are or where they graduated and we have no hint of whether other salient aspects have limited their choices) - something that is not usually noted in the literature.

Some aspects are noted but not fully integrated - of course the respondents are older than new local graduates (line 98) - they have not only done similar quals but they also have work experience that they bring in and they have had to jump through hoops to get taken seriously. This implies an element of ageism and perhaps racism - but the key problem is that immigration policies and practices have misled them and the medical registration system is not integrated. Just looking at the range of dates of the references leaves an impression nothing has changed for decades.  Having said that, much of this study is limited by the recognised issues with the sampling (lines 167-173) - so getting much further forward in understanding is necessarily limited.

Line 222 - it might be worth noting the pathways that are closed to people wanting to follow alternative careers - for example (not necessarily true of Canada - you will need to check) universities will not permit enrolment in a medical course by anyone who has already graduated elsewhere with a medical degree even if that degree is not recognised by the medical councils - some of these closed pathways may affect those who have not yet actively moved to alternative career paths - in other words a significant portion of the population of interest required to understand the nuances of the topic of this paper has been excluded.

Line 370 - facility in English is likely to be a major component in gaining registration through alternative paths (such as the US certification system) and it is clear from the citations that English is a factor - although time and cost is cited frequently as central.

Line 421 says "interested in" but this group has been specifically excluded in the survey design (though line 484 acknowledges that this would be a good extension)

Line 504 - "should be supported" - a vitally important point because a core characteristic of self-driven migrations is that they are not lacking initiative (they have demonstrated get-up-and-go) or desire - it is the lack of support that stands out.

Figure 1: think about whether "intuitive" is the best label for the model - the other labels are also a bit ill-thought-out but "intuitive" jumps out as inadequate.

Table 1: Over half of the sample are from South Asia or Nigeria - with a group from Philippines (which traditionally exports many highly qualified people who are in nursing and geriatric care fields but unable to register as physicians). Is it worth mentioning that this may be the result of how the sample was drawn or is it to do with immigration policies?  The migrants' status implies that most have been in Canada for an extended period.  The gender balance may also be significant - especially if there are constraints on males moving into nursing for example.  Is it possible to tease out some cross-tabs to look at inter-relationships between the variables? Just feels like a story half-told - though the issue of confidentiality and privacy may limit the options here.

The paper as it stands looks rushed. In particular I would at least expect academic rigour in the citations - several are incomplete (lines 531-2, 592-3, 623 just as 3 of several examples) and line 647 is not only incomplete but also incorrectly formatted.

The English needs an edit - also check that each of the direct quotations of respondents are absolutely as recorded.  

Author Response

Comments from Reviewer 3

  1. This is a paper on an extremely important topic that applies to a number of countries like Canada - the problem appears to be that medical qualifications and registration systems that are not integrated with migration policies - a systemic problem because the medical people insist it is about patient safety and not elitist views of the local qualifications, whereas the immigration policy folks place emphasis on level scores rather than content - an example of two silos failing to talk to each other. Useful point: Canadian-born/foreign educated have the same problem (albeit with a sample of only 2 people and from the paper we do not know whether they are male or female or what their specialities are or where they graduated and we have no hint of whether other salient aspects have limited their choices) - something that is not usually noted in the literature.

Reply:

Thank you for your recognition of the importance of the topic and your valuable comment on the disintegrated systems of medical qualifications and registration systems and the migration policies.

  1. Some aspects are noted but not fully integrated - of course the respondents are older than new local graduates (line 98) - they have not only done similar quals but they also have work experience that they bring in and they have had to jump through hoops to get taken seriously. This implies an element of ageism and perhaps racism - but the key problem is that immigration policies and practices have misled them and the medical registration system is not integrated. Just looking at the range of dates of the references leaves an impression nothing has changed for decades. Having said that, much of this study is limited by the recognised issues with the sampling (lines 167-173) - so getting much further forward in understanding is necessarily limited.

Reply:

Thank you for this constructive comment. We agree with you the elements of discriminations affecting this issue. We have incorporated that in the discussion of the revised manuscript.

Please see page 11, lines 418-422

Non-recognition of immigrants’ education and work experience has been well documented in the literature, resulting in limited access to opportunities IEPs can pursue with interest, passion, and dignity [42]. These experiences were also shared by our participants and other studies and deemed as systemic discrimination including ageism and racism [43,44]

Also, yes; the sampling was purposeful in this study to ensure we obtain the answers to our research question, thus the findings of this study is limited and would be extrapolating to inform the issue you pointed out. We also acknowledge the sampling issue in the limitation section.

Please see page 12, lines 474-489

The moderator of the focus groups ensured that all participants contribute to each idea to reduce bias in our findings due to dominant participants. Despite the lack of interpersonal interactions conducting focus groups online came out as rather beneficial for this study as the participants could express themselves more freely, and could join from their homes anywhere in Canada [55]. However, most of the participants in this study were from Alberta (71.4%) due to our purposive sampling technique. We wanted to make sure that the participants were genuinely working in or considering alternative careers, which was more feasible for us from Alberta compared to other provinces. Further, perhaps participants from Alberta felt more interested than those from other provinces in participating in research conducted by the local and familiar university and community organizations. As IEPs across Canada encounter similar struggles in achieving their primary career and have similar educational and socio-cultural backgrounds, we believe the findings of our study can be applicable to a great extent in other provinces. We also observed a higher proportion of female participants in our study. We acknowledge this might be a limitation due to our chosen non-random sampling technique; however, other population-based studies also found a higher proportion of female participants in their studies [11,12]. 

  1. Line 222 - it might be worth noting the pathways that are closed to people wanting to follow alternative careers - for example (not necessarily true of Canada - you will need to check) universities will not permit enrolment in a medical course by anyone who has already graduated elsewhere with a medical degree even if that degree is not recognised by the medical councils - some of these closed pathways may affect those who have not yet actively moved to alternative career paths - in other words a significant portion of the population of interest required to understand the nuances of the topic of this paper has been excluded.

Reply:

Thank you for bringing these thoughtful aspects. From our experience in the past years, we can say probably in Canadian situation, the enrollment to the medical courses/other courses are not restricted for people with foreign medical degrees. We came across many such examples being enrolled in many programs as well as even medical school again. We believe bringing that aspect might be a little bit out of context for this study, especially in the Canadian perspectives. That’s why we opted for not to comment on this issue in the revised manuscript.

  1. Line 370 - facility in English is likely to be a major component in gaining registration through alternative paths (such as the US certification system) and it is clear from the citations that English is a factor - although time and cost is cited frequently as central.

Reply:
Thank you. We agree with your observations. We also discussed that in the discussion in the revised manuscript.

Please see page 11, lines 424-428

These decisions to update skills including meeting certain score requirements in English proficiency tests often require immense resources, such as time and money, leading to a disproportionate rate of unemployment, underemployment, and lower earning for IEPs as compared to the native Canadian professional population [46].

  1. Line 421 says "interested in" but this group has been specifically excluded in the survey design (though line 484 acknowledges that this would be a good extension)

Reply:

Thank you for your comment. Indeed, we also believe that this would have been a good extension. We are in the process of conducting future studies in this aspect.

  1. Line 504 - "should be supported" - a vitally important point because a core characteristic of self-driven migrations is that they are not lacking initiative (they have demonstrated get-up-and-go) or desire - it is the lack of support that stands out.

Reply:

Thank you for your comment. Yes. We also believe so.

  1. Figure 1: think about whether "intuitive" is the best label for the model - the other labels are also a bit ill-thought-out but "intuitive" jumps out as inadequate.

Reply:

Thank you for the comment. We used this labels and the concept from Singh & Greenhaus’s article (reference no 54). They compared three career decision-making strategies for employees going through a career change that included rational (based on careful thought), intuitive (based on emotional satisfaction), and dependent (assistance/approval from others) strategies. We find this concept is useful to explain the factors we identified. And keeping ture to the original framework proposed by the authors (Singh & Greenhaus), we opted to use their terms and cited their paper appropriately . 

  1. Table 1: Over half of the sample are from South Asia or Nigeria - with a group from Philippines (which traditionally exports many highly qualified people who are in nursing and geriatric care fields but unable to register as physicians). Is it worth mentioning that this may be the result of how the sample was drawn or is it to do with immigration policies? The migrants' status implies that most have been in Canada for an extended period. The gender balance may also be significant - especially if there are constraints on males moving into nursing for example.  Is it possible to tease out some cross-tabs to look at inter-relationships between the variables? Just feels like a story half-told - though the issue of confidentiality and privacy may limit the options here.

Reply:

We openly invited to participate in the studies. However, we only included who fulfilled the inclusion criteria. As it was a qualitative study, we are not sure that we would be able to infer that explanation regarding the country of origins. However, we acknowledged the gender imbalance in the discussion section.

Please see page 12, lines 486-489

We also observed a higher proportion of female participants in our study. We acknowledge this might be a limitation due to our chosen non-random sampling technique; however, other population-based studies also found a higher proportion of female participants in their studies [11,12].

And, yes, we agree that in addition to the limitation of qualitative study this cross-tabbing will may compromise confidentiality and privacy issues.

  1. The paper as it stands looks rushed. In particular I would at least expect academic rigour in the citations - several are incomplete (lines 531-2, 592-3, 623 just as 3 of several examples) and line 647 is not only incomplete but also incorrectly formatted.

Reply:

Thank you for the comment.

We believe that the issue of rigour in the citation have arisen from the issues with the journal’s auto-conversion of our paper after our submission.

We have provided a very standard version (double-spaced 12 Times New Roman and properly cited/formatted references). The journal submission portal converted our submitted manuscript to the current form. The conversion may affect the formatting and citations.

Also, for your kind note, this topic is a less explored are. Thus, much information is from the Grey literature that don’t have a full citation as a journal published articles.

We formatted the reference list in the revised manuscript.

  1. The English needs an edit - also check that each of the direct quotations of respondents are absolutely as recorded.

Reply:

Thank you. Yes. We have used direct quotations of the participants absolutely as recorded.